# Functional Characterization of Aluminum (Al)-Responsive Membrane-Bound NAC Transcription Factors in Soybean Roots

**DOI:** 10.3390/ijms222312854

**Published:** 2021-11-27

**Authors:** Yan Lin, Guoxuan Liu, Yingbing Xue, Xueqiong Guo, Jikai Luo, Yaoliang Pan, Kang Chen, Jiang Tian, Cuiyue Liang

**Affiliations:** 1Root Biology Center, State Key Laboratory for Conservation and Utilization of Subtropical Agro-Bioresources, South China Agricultural University, Guangzhou 510642, China; ly617590089@126.com (Y.L.); liuguoxuanscau@163.com (G.L.); g13095328032@126.com (X.G.); scauljk@126.com (J.L.); 18320353151@163.com (Y.P.); chenkang@scau.edu.cn (K.C.); jtian@scau.edu.cn (J.T.); 2Department of Resources and Environmental Sciences, College of Chemistry and Environment, Guangdong Ocean University, Zhanjiang 524088, China; yingbinxue@yeah.net

**Keywords:** aluminum resistance, soybean, NTLs, arabidopsis, RNA-seq assay

## Abstract

The membrane-bound NAC transcription (NTL) factors have been demonstrated to participate in the regulation of plant development and the responses to multiple environmental stresses. This study is aimed to functionally characterize soybean NTL transcription factors in response to Al-toxicity, which is largely uncharacterized. The qRT-PCR assays in the present study found that thirteen out of fifteen *GmNTL* genes in the soybean genome were up-regulated by Al toxicity. However, among the Al-up-regulated *GmNTL*s selected from six duplicate gene pairs, only overexpressing *GmNTL1*, *GmNTL4*, and *GmNTL10* could confer Arabidopsis Al resistance. Further comprehensive functional characterization of *GmNTL4* showed that the expression of this gene in response to Al stress depended on root tissues, as well as the Al concentration and period of Al treatment. Overexpression of *GmNTL4* conferred Al tolerance of transgenic Arabidopsis in long-term (48 and 72 h) Al treatments. Moreover, RNA-seq assay identified 517 DEGs regulated by GmNTL4 in Arabidopsis responsive to Al stress, which included *MATE*s, *ALMT*s, *PME*s, and *XTH*s. These results suggest that the function of *GmNTL*s in Al responses is divergent, and *GmNTL4* might confer Al resistance partially by regulating the expression of genes involved in organic acid efflux and cell wall modification.

## 1. Introduction

In soils with pH values below 5.0, Aluminum (Al) is solubilized as highly phytotoxic Al^3+^, which is able to interact with various cell components, leading to the disruption of signal transduction, the plasma membrane, the cytoskeleton, and DNA in the nuclei [1]. The damages in root tips ultimately inhibit root growth, along with the uptake of water and nutrients, and result in the loss of crop yield [2]. Since approximately 30% of the world’s total land area and over 50% of the world’s potentially arable lands are acidic, Al toxicity is considered a major limiting factor for crop production worldwide [2,3].

Plants have evolved diverse Al resistance mechanisms, which could be divided into two distinct categories. One is Al^3+^ exclusion from the root apex, while the other allows plants to tolerate internal Al^3+^ in the symplast [3]. The Al^3+^ exclusion mechanism is mediated mainly by the exudation of Al-chelating organic acids from root tips, which form stable chelating complexes with Al^3+^ in the rhizosphere and prevent its uptake into root cells. For instance, in Arabidopsis (*Arabidopsis thaliana*), the Al-responded accumulation of AtALMT1 (Al-activated Malate Transporter) and AtMATE (Multidrug and Toxic Compound Extrusion) transporters enhance the secretion of malate and citrate, limiting the Al^3+^ uptake and conferring plant Al resistance [4,5,6]. While the organic acids exudates from roots prevent Al up-taking into the root symplast, the cell walls also compete with the organic acids for Al binding sites [7]. The cell wall acts as the first biological interface in contact with Al. The binding of Al in the cell wall impairs the function of both apoplastic and symplastic [2]. Thus, plants involve a serial of cell wall modifications to reduce the Al bindings. For instance, under Al toxicity conditions, the expressions of four Arabidopsis xyloglucan endotransglucosylase/hydrolase (XTH) genes, including *XTH14*, *15*, *17*, and *31*, were significantly down-regulated [8,9]. Among them, the lost function of both *XTH17*and *XTH31* reduced the accumulation of cell wall xyloglucan and the Al binding capacity in Arabidopsis, suggesting that *XTHs* play important roles in Al resistance [9,10]. Pectins are another group of complex polysaccharides being targeted by Al [11]. The methylation degree of pectin, which is controlled by the activity of pectin methylesterase (PME) enzymes (EC 3.1.1.11), showed a negative correlation with the amount of Al binding to the pectin matrix [7,12,13]. Increasing experimental results have provided evidence that the PME-mediated pectin demethylation intensifies Al accumulation and Al-induced inhibition of root elongation [14,15,16]. Therefore, understanding the molecular regulatory pathways of the genes involved in organic acid exudation, as well as modification of cell wall components (e.g., pectin and hemicellulose), would largely expand the understanding of plant Al tolerance mechanisms.

The mechanisms underlying Al tolerance are regulated by a range of transcription factors, such as those belonging to *STOP1*, *WRKY*, *MADS*, and *NAC* super-families [17,18,19,20,21,22]. Among them, the NAC supper-family has been specifically identified in plants; proximately, 110 *NAC* genes in the model Arabidopsis, 150 in rice, and 180 in the soybean genome [23,24]. The NAC transcription factors have been reported to involve plant Al responses. For example, the expression of 25 rice *OsNAC* genes were significantly regulated by Al treatment, some of which were also responsive to other environmental stresses [25,26]. The results suggested that characterization of the NAC transcription factors would be an effective tool for the development of broad-spectrum stress-tolerant crop plants [25,27]. In the NAC super-family, there is a subfamily called “membrane-bound NAC transcription factors” (NTLs) [27]. In the Arabidopsis genome, there are 13 *NTL* genes, which are dispersed in the I, IV, and VII sub-group of NAC super-family [28]. NTL proteins contain a conserved NAC domain in the N-terminal region as other NAC members, and they also have a distinguishing C-terminal transmembrane (TM) motif in their C-terminal region. It has been suggested that the NTL proteins are released from the membranes by proteolytic cleavage and subsequently transported into the nucleus to regulate the downstream genes in response to the specific developmental or environmental cues [29,30,31,32,33,34,35,36,37,38,39,40,41,42,43,44,45]. Since the first *NTL* gene (*NTM1*), which mediated cytokinin signaling during cell division, was found in Arabidopsis in 2006 [32], the biological functions of NTL homologs have been studied in a variety of plants. It has demonstrated that *NTL7* (i.e., RAO2/ANAC017) was indispensable for oxidative stress [35]. Unlike NTL7, NTL8 was shown to negatively regulate seed germination in response to salt stress [36]. In addition to salt and oxidative stress, other factors such as heat, high-light, osmotic, and drought stress were also reported affecting NTLs activities, which functioned in regulating leaf development and senescence, flavonoid biosynthesis, and flowering [37,38,39,40,41,42,43,44]. Besides abiotic stress, *SlNACMTF3* and *SlNACMTF 8* from the tomato (*Solanum lycopersicum*) *NTL* family were shown to participate in biotic stress response [45]. These findings suggest that different *NTL* members have very diverse functions. However, little is known about whether NTLs are also involved in plant responses to Al toxicity.

Soybean (*Glycine max*) is an important crop providing proteins and oils worldwide. However, Al toxicity strongly limits soybean production in tropic and subtropic areas. In soybean, although several genes, including the organic acid transporter genes (e.g., *GmALMT1* and *GmMATE*s), the glycine-rich protein-like gene (e.g., *GmGRPL*) and the IRON REGULATED/ferroportin family gene (e.g., *GmIREG3*) have been found conferring Al toxicity [46], the gene resources involved in soybean Al tolerance remain to be expanded. A previous comparative transcriptome analysis provided preliminary clues that the expression of some soybean *NTL* genes is significantly up-regulated by Al toxicity [47]. In the soybean genome, there is a total of 15 *GmNTL* members, which are divergent at the transcriptional, posttranscriptional, and protein levels [48]. Among these 15 *GmNTL*s, only *GmNTL1* and *GmNTL11* have been functionally characterized as being involved in plant H_2_O_2_ resistance [48]. Therefore, to explore the possible role of *GmNTLs* in Al tolerance, the present study first performed quantitative reverse-transcription PCR (qRT-PCR) to investigate the Al-responsive expression pattern of all 15 *GmNTL*s in soybean root tips. Furthermore, six Al-up-regulated *GmNTL*s from six duplicate gene pairs were overexpressed in Arabidopsis to compare their function in Al resistance. The regulatory mechanism of *GmNTL4* in Al tolerance was further analyzed by RNA-seq. Our results suggest that soybean *GmNTL* members have a divergent function in Al tolerance, and GmNTL4 might confer Al resistance partially by regulating genes that are involved in organic acid efflux and cell wall modification.

## 2. Results

### 2.1. Expression Profiles of GmNTLs Regulated by Al Toxicity

A total of 15 GmNTL members were identified in the soybean genome and named GmNTL1-GmNTL15 according to their positions on chromosomes (Appendix A, [48]). The predicted molecular weight of the GmNTL proteins ranged from 5.0 kDa to 7.7 kDa, and the predicted isoelectric point of most GmNTLs members ranged from 4.5 to 6.0, except for GmNTL9 and GmNTL10, which were higher than 7.0 (Appendix A). Phylogenetic tree analysis classified these 15 GmNTLs into four subgroups and formed seven duplicate gene pairs. The subgroup I included GmNTL2/12 and GmNTL8/13, while subgroup II contained GmNTL6/15 and GmNTL7/14. The other five members, including duplicate pairs GmNTL1/11 and GmNTL3/4, as well as GmNTL5, belonged to subgroup III, while GmNTL9/10 alone was in subgroup IV (Figure 1A).

To determine the regulation of GmNTLs in response to Al toxicity on soybean root tips, three-day-old soybean seedlings were treated with Al for 48 h, when the primary root growth was significantly inhibited by Al treatment (Figure 1B). The 0–2 cm root tips were subsequently harvested and used for qRT-PCR to examine the expression levels of GmNTLs. Results showed that except for GmNTL9 and GmNTL13, the transcription levels of the other GmNTLs genes in soybean root tips were up-regulated in different degrees by Al toxicity, ranging 1.5–9 fold higher than the control. For instance, the expression level of GmNTL11, GmNTL4, and GmNTL15 were increased by about 1.5, 2, and 9 times compared to the control (Figure 1C).

### 2.2. Subcellular Localization of GmNTLs

To ascertain the function of GmNTLs as transcriptional factors, five GmNTLs members, including GmNTL1, GmNTL2, GmNTL4, GmNTL7, and GmNTL10, were selected for subcellular localization analysis by transiently expressed in tobacco (*Nicotiana benthamiana*) leaves. The fluorescence of GFP-GmNTLs was observed at 48 h after transformation. Results showed that in the tobacco leaf epidermic cells, the green fluorescence derived from most of the GFP-GmNTLs, except for GmNTL10, were found clearly in both the nucleus and the cell periphery, which merged well with the red fluorescence derived from AtPIP2A-mCherry for the indication of the plasma membrane. While the green fluorescence derived from GFP-GmNTL10 was found in the plasma membrane, nucleus, and cytoplasm (Figure 2). Moreover, at 72 h after transformation, the green fluorescence of GFP-GmNTL4 was only observed in the nucleus (Appendix A). This result suggested that the selected GmNTLs were able to localize in both membrane and nucleus, and the localization of GmNTLs is changeable and dependent on environmental stimuli.

### 2.3. Functional Analysis of GmNTLs in Response to Al Toxicity

Six genes, including *GmNTL1*, *GmNTL2*, *GmNTL4*, *GmNTL7*, *GmNTL8*, and *GmNTL10* from six duplicate gene pairs, were selected for functional characterization by overexpressing in Arabidopsis. It showed that though the expression of these *GmNTL*s were all up-regulated by Al in soybean root tips, not all of them were able to confer the Al tolerance in Arabidopsis. Among them, only overexpression of *GmNTL1*, *GmNTL4*, and *GmNTL10* significantly increased the Arabidopsis root elongation in response to Al toxicity (Figure 3). The root elongation of two *GmNTL4*-overexpressing lines were about 1.54 and 1.60 folds (*p* < 0.001) higher than that of the wild type, respectively. While root elongation of the *GmNTL1* and *GmNTL10* overexpressing lines were about 1.46–1.78 and 1.20–1.36 fold higher than in the wild type, respectively (Figure 3).

### 2.4. Functional Characterization of GmNTL4 in Response to Al Toxicity

Based on the results, *GmNTL4* was chosen for further functional characterization. The expression pattern of *GmNTL4* was examined in different root segments in response to various Al treatments. It showed that under both low (25 μM) and moderate (50 μM) Al concentrations, the expression of *GmNTL4* in 0–2 cm root tips was increased with the Al-treatment time prolonging. For example, the expression level of *GmNTL4* under 50 μM Al condition was 1.8, 6.9, and 21.1 fold higher than the control (0 μM Al) at 24, 48, and 72 h after Al treatment. Whereas, the expression of *GmNTL4* under high Al concentration (100 μM) remained at almost the same level after it was enhanced by 24 h of Al treatment (Figure 4A). In addition, *GmNTL4* in different root segments responded differentially to Al toxicity. Compared to the control (0 μM Al), the expression of *GmNTL4* under Al treatment was 4.9 and 2.9 fold higher in 0–2 and 2–4 cm root fragments, respectively. However, in the >4 cm root segment, no significant difference was observed in the *GmNTL4* expression between control and Al treatment (Figure 4B). These results suggested that the expression of *GmNTL4* in response to Al stress depends on root tissues, as well as the Al concentration and period of Al treatment.

Furthermore, the function of *GmNTL4* in Al tolerance was examined, crossing a wide time frame from 24 to 72 h. Results showed that under short-term Al treatment (24 h), no significant differences in root elongation between wild-type plants and *GmNTL4-*overexpressing plants were observed (Figure 4C). However, under prolonging Al treatment (48 and 72 h), *GmNTL4*-overexpressing plants showed higher Al-resistance than wild-type plants, as indicated by the higher primary root elongation of *GmNTL4*-overexpressing lines. For instance, the root elongation of the *GmNTL4*-overexpression lines OX1 and OX2 was 1.56 and 1.68 folds higher than that of the wild type at 72 h after Al treatment, respectively (Figure 4C).

### 2.5. Gene Expression Profiles Affected by GmNTL4 Overexpression in Arabidopsis in Response to Al-Toxicity

The finding that GmNTL4 confers Arabidopsis Al tolerance prompted us to investigate how GmNTL4 regulates downstream genes expression. The roots of *GmNTL4*-overexpression and wide-type plants were freshly harvested for RNA-seq analysis. The result showed that under Al conditions, 213 and 304 genes were up- or down-regulated in the *GmNTL4*-overexpressing plants, respectively (Figure 5A). KEGG and Gene Ontology (GO) analysis revealed that these differentially expressed genes (DEG) are involved in various biological processes, such as zeatin biosynthesis, pentose phosphate pathway, nitrogen metabolism, glyoxylate, and dicarboxylate metabolism, etc. (Figure 5B and Appendix A).

Moreover, the RNA-seq results showed that GmNTL4 either positively or negatively regulates genes involved in organic acid transportation and cell wall modification (Figure 5C), which are closely related to Al tolerance. For example, two *MATE* genes *DTX14* (*At1g71140)* and *DTX15 (At2g34360*), were significantly up-regulated in *GmNTL4*-overexpressing roots, while another *MATE* gene (*At1g15180*) and the *ALMT12* (*At4g17970*) were down-regulated. Among the genes involved in cell wall modification, the expression of one polygalacturonase gene *DTX13* (*At2g33160*) and one galactinol synthase gene *GOLS4* (*At1g60470*) were up-regulated in *GmNTL4*-overexpressing roots with 3.72 and 4.02 fold higher than that of the wild type. The expressions of the other genes, including three *PME* genes *PME60* (*At5g51500*), *PME59* (*At5g51490*), *PME25* (*At3g10720*), two polygalacturonase gene (*At2g43890*, *At2g43880*), two cellulose synthase genes *CSLB5* (*At4g15290*) and *CSLB3* (*At2g32530*), two xyloglucan endotransglucosylase genes *XTH17* (*At1g65310*) and *XTH33* (*At1g10550*), and one pectin lyase gene (*At3g59850*) were decreased in the *GmNTL4*-overexpressing roots (Figure 5C).

To further confirm the RNA-seq results, qRT-PCR was performed to evaluate the expression pattern of several selected genes in response to Al stress using freshly harvested Arabidopsis roots treated with and without Al. Results showed that similar to the data obtained by RNA-seq analysis, the expression of *GOLS4* (*At1g60470*) and *At2g33160* were up-regulated, while the expression of *ALMT12* (*At4g17970*), *XTH33* (*At1g10550*), *PME59* (*At5g5149*0), *CSLB5* (*At4g15290*), *PMDH2* (*At5g09660*), *At2g43880*, and *At2g43890* were down-regulated by Al (Figure 5D). Moreover, several putative NAC binding CATGT motifs were found in the promoter regions of these genes regulated by GmNTL4 in Arabidopsis (Appendix A). These results indicate that GmNTL4 might confer the Al resistance in Arabidopsis partially by regulating both the organic acid exudation and cell wall modification.

## 3. Discussion

As a subfamily of NAC, NTL transcription factors are recognized playing crucial regulatory functions [29,30,49]. In the model plant Arabidopsis, the regulatory function of AtNTL1 and AtNTL7 in oxidative stress, AtNTL4 in heat stress, AtNTL6 in cold stress, AtNTL8 in salt stress, AtNTL9 in osmotic stress, AtNTL11 in high-light stress have been reported [35,36,37,38,39,40,41,42,43,44]. In soybean, the function of GmNTL1 and GmNTL11 in plant adaptation to salt and drought stresses have also been well characterized [48]. Although the function of NTLs in plant development and resistance to environmental stresses have been widely studied, little attention has been paid to consider the involvement of NTLs in plant Al responses. Therefore, with *GmNTL*s as candidate genes, the present study tried to find out the possible regulatory functions of NTL transcription factors in plant Al tolerance.

Our qRT-PCR data showed that the expression of 13 out of 15 *GmNTLs* was significantly enhanced by 48 h of Al treatment (Figure 1C). These results suggested that *GmNTL* might be involved in soybean Al adaptive responses. Evidence from a comparative transcriptome analysis provided strong supports to this hypothesis by finding that the expression of *GmNTL5* and *GmNTL15* in the root tips of soybean genotype M90-24 was significantly up-regulated by Al treatment [47]. When we look deep insight the Al-response expression pattern of *GmNTL4*, we found that low Al concentration (25 μM) did not, but high Al concentration (100 μM) did enhance the expression of *GmNTL4* after 12 h of Al treatment, and the Al-enhanced expression of this genes was showed in the root tips, but not in the segments near to the root-base (Figure 4). It thus suggested that the Al-regulated *GmNTL* genes’ expressions are due to the period of Al toxicity and the Al concentration in the growth medium. Based on these results, we can explain why some previous transcriptomic analyses did not identify any Al-responded *NTL* genes in other plants, such as maize [50]. Moreover, it is noticed that the Al-induced *GmNTL* genes included the previous functionally characterized *GmNTL1* and *GmNTL11*, which have been reported to be induced by H_2_O_2_, PEG, salt, and cold treatments [48]. It thus suggests Al toxicity is not a specific stimulus for the regulation of *GmNTL*s expression; instead, transcription of *GmNTL*s could be regulated by multiple environmental stimuli. Similarly, some other *NTL* genes were also reported to be regulated by several environmental stresses. For instance, the expression of *AtNTL6* was regulated by both pathogenesis and abiotic stresses, including the salinity, coldness, and drought stresses [33,40]. Therefore, multiple-stresses-regulated expression patterns seem to be a common occurrence among plant *NTL* genes. Overall, the expression pattern of *GmNTLs* in response to Al toxicity indicated that *GmNTLs* genes could be involved in soybean response to Al toxicity.

To further verify the function of *GmNTL*s in plant Al resistance, we chose six *GmNTL* genes from six duplicate gene pairs and overexpressed them in Arabidopsis (Figure 1A and Figure 4). Although the expression of all these genes was significantly up-regulated by Al stress in soybean root tips, not all of them were able to confer Al tolerance in Arabidopsis. (Figure 4). Two possible reasons could explain these functional divergences among soybean *GmNTL* members. One is that the functions of *GmNTL*s are indeed genetically different; the other might be attributed to the false subcellular localization of GmNTLs in transgenic Arabidopsis. For previous studies found that when the full-length *NTL*s were overexpressed in plants, often no phenotypic differences were observed, whereas overexpressing *NTL*s without the transmembrane motif would show a corresponding phenotypic difference between wild type and transgenic seedlings [41,48,51]. It thus proposes. It thus proposes that the release of NTL proteins from the membrane into the nucleus determined their transcription regulation functions. Therefore, a possible reason why overexpression of *GmNTL2*, *GmNTL7*, and *GmNTL8* did not lead to distinct phenotypes could be due to their improper membrane release in transgenic Arabidopsis seedlings.

To understand the possible regulatory mechanisms of GmNTLs under Al stress, we chose *GmNTL4* as the candidate gene for further functional analysis. Results showed that overexpression of *GmNTL4* could significantly enhance the Al resistance in Arabidopsis, especially under prolonging Al treatment (Figure 4C). RNA-seq was further conducted to identify the downstream genes of *GmNTL4* in Arabidopsis and found a total of 517 genes were differentially expressed in the *GmNTL4*-overexpression plants under Al toxicity conditions. KEGG analysis revealed that these genes are involved in various biological processes. Among these DEGs, two MATE genes (*At1g71140* and *At2g34360*) were significantly up-regulated by overexpression of *GmNTL4*. MATE transporters have been widely reported in many plant species mediating the citrate exudation in response to Al toxicity [5,52,53,54,55,56]. In Arabidopsis, *AtMATE* is expressed in the mature portions of the roots and confers Al tolerance independently with *AtALMT1* [5,52]. Another Arabidopsis MATE transporter (AtDTX30) promotes Al tolerance by modulating auxin levels in the root tips [57]. Except for these two, the function of other Arabidopsis MATE transporters in Al tolerance has not been reported. Nevertheless, the up-regulated expression of two *MATE*s in the *GmNTL4*-overexpressing Arabidopsis roots indicates a possibility that GmNTL4 might confer Al tolerance by regulating the expression of some MATE transporters in Arabidopsis. In addition, we also found that the expression of *AtALMT1*2 (*At4g17970*) was significantly reduced in the *GmNTL4*-overexpression plants. *AtALMT12* has been reported highly expressed in both root pericycle and leaf guard cells [58]. It is clear that AtALMT12 was involved in stomatal closure, affecting the plant photosynthesis and drought tolerance [59,60,61], but no studies have been conducted to characterize the function of AtALMT12 in Al tolerance. Therefore, whether the GmNTL4-mediated Al tolerance involved AtALMT12 remains to be clarified.

We also found two xyloglucan endotransglucosylase genes *At1g65310* and *At1g10550*, which have been designated as *XTH17* and *XTH33*, respectively [62], were significantly down-regulated by the overexpression of *GmNTL4*. The previous study showed that the expression of *XTH17* was substantially reduced in the presence of Al [9]. Moreover, mutation of *XTH17* showed lower XET action and hemicellulose content, retained less Al, but was more Al resistant than the wild type [9]. The results provide important evidence for our hypothesis that *GmNTL4* confers the Al tolerance in Arabidopsis by suppressing the expression of *XTH17*. In addition, three pectin methylesterase (PME) encoding genes, including *At3g10720*, *At5g51490*, and *At5g51500*, were all down-regulated. Studies in a range of plant species have demonstrated that Al sensitivity largely depends on the degree of PME activity [63]. For instance, in rice, maize, and pea (*Pisum sativum*), the Al-sensitive cultivars showed higher pectin methylesterase (PME) activities and a higher portion of demethylated pectin than that in the Al-resistant cultivars [7,12,64]. The PME activity in the cell wall is largely associated with the transcription of *PME* genes. Transcriptional analysis in maize revealed that Al exposure caused significantly higher expression of a *PME* gene (MZ00000091) in the Al-sensitive genotypes [65]. Moreover, transgenic rice overexpressing *OsPME14* exhibited higher PME activity and Al content in the root tip cell wall and became more sensitive to Al stress [16]. Thus, the down-regulation of the PME genes in *GmNTL4*-overexpressing plants might be another reason for their higher Al resistance. We further searched the promoter regions of the genes regulated by GmNTL4 and found more or less putative NAC-binding CATGT motifs in all these genes’ promoter regions (Appendix A). Therefore, these results indicate that GmNTL4 might confer Al tolerance partially by regulating the expression of multiple genes involved in organic acid efflux and cell wall modifications.

In conclusion, here, the expression pattern of all fifteen soybean *GmNTL* genes in response to Al toxicity was evaluated in soybean root tips, and the divergence of *GmNTL*s’ function in Al resistance were determined by heterologously overexpressing six *GmNTL* genes in Arabidopsis. The findings suggest that, like other abiotic stress, Al is able to stimulate the expression of most *GmNTL*s and the regulatory function of GmNTLs in Al resistance was divergent. Moreover, GmNTL4 might confer Al resistance partially by regulating the expression of genes involved in organic acid efflux and cell wall modification.

## 4. Materials and Methods

### 4.1. Plant Material and Growth Conditions

Soybean genotype YC03-3 was chosen for the expression pattern examination of *GmNTLs*. Uniform soybean seeds were germinated in paper rolls soaked with 1/4 Hogland nutrient solution. Four days after germination, uniform seedlings were selected and scanned to record the initial root length and subsequently treated with or without 50 µM AlCl_3_ in 0.5 mM CaCl_2_ solution (pH 4.2) for 48 h. Root tips (0–2 cm) were separately harvested for gene expression assays, and root length was measured to calculate the root elongation. All experiments had four biological replicates, with 8 plants in each replicate.

### 4.2. Identification of the GmNTL Members in Soybean Genome

According to the reports by Li et al. [48], BLAST searches were performed using all identified GmNTLs protein sequences published at the Phytozome website (https://phytozome.jgi.doe.gov/pz/portal.html, accessed on 21 May 2019). Basic information for each identified GmNTLs member (e.g., numbers of exons and introns, length of ORF) was extracted from the same website. Isoelectric points and molecular protein weights were predicted using the ExPASy web server (http://www.expasy.org/, accessed on 30 May 2020). Multiple sequence alignment of the GmNTL proteins was conducted using the neighbor-joining method with 1000 bootstrap replicates in MEGA 7.0, as described previously [66]. Conserved motifs were predicted by MEME web server (http://meme-suite.org/, accessed on 13 July 2021) and TBtools (http://www.tbtools.com/, accessed on 13 July 2021) was used to graphic the phylogenetic tree with conserved motifs.

### 4.3. RNA Extraction and Quantitative Real-Time PCR (qRT-PCR) Analysis

Total RNA was extracted from various plant tissues using an RNA-solve reagent (OMEGA Bio-Tek, Norcross, GA, USA) following the manufacturer’s instructions. The RNA samples were added with RNase-free DNase I (Invitrogen, Carlsbad, CA, USA) to remove genomic DNA. The first complementary DNA was synthesized from 1 µg RNA by using GoScript (Promega, Madison, WI, USA), according to the manufacturer’s instructions. The qRT-PCR analysis was conducted using an ABI7500 real-time PCR system (Thermo Fisher Scientific, Waltham, MA, USA). Each reaction contained 2 μL of 1:20 diluted reverse transcription product, 0.4 μL of forward and reverse primers (0.2 μM final concentration), 7.2 of μL DNase-Free ddH_2_O and 10 μL of 2 × SYBR™ Green PCR master mix (Thermo Fisher Scientific, Waltham, MA, USA). The PCR cycling system consisted of incubation at 95 °C for 1 min and followed by 40 cycles of 95 °C for 15 s, 60 °C for 60 s and 72 °C for 30 s. The dissociation curve was obtained after the last cycle. For target genes expression analysis, the specific primer pairs were designed as listed in Appendix A. Expression levels of the soybean housekeeping gene *GmEF1-α* (Glyma.17G186600) and the Arabidopsis housekeeping gene *AtEF-α* (At5G60390) was used to normalize the samples, as described previously [67].

### 4.4. Subcellular Localization of GmNTLs

To determine the subcellular localization of GmNTLs, the full-length cDNA of *GmNTL*s was amplified with the specific primer pairs (Appendix A), and cloned into the *pEGAD* vector with green fluorescent protein (GFP) fused with their N-terminus to produce *35S::GFP-GmNTL*s constructs. The *35S::GFP* and the *35S::GFP-GmNTL*s constructs were separately introduced into *Agrobacterium tumefaciens* GV3101 and then transformed into tobacco (*Nicotiana benthamiana*) leaves. The plasma membrane marker plasmid of *AtPIP2A-mCherry* was co-transformed with the *35S::GFP* and 35S::*GFP-GmNTL*s constructs for co-localization analysis. The blue-fluorescent DNA staining chemical 4’,6-diamidino-2-phenylindole (DAPI) was used to stain the nucleus. The green fluorescence derived from GFP was observed by the confocal scanning microscope at 488 nm excitation/507 nm emission (Zeiss LSM780, Oberkochen, Germany).

### 4.5. Effects of GmNTLs Over-Expression on Arabidopsis Al Resistance

The recombinant plasmid of *35S::GmNTL*s was introduced into the *Agrobacterium tumefaciens* strain GV3101 and then transformed into Arabidopsis by the floral dip method [68]. The T1 generation transgenic Arabidopsis seeds were cultured in MS medium containing herbicide for 14 days. After that, the surviving seedlings were transplanted into peat soils for another 7 days, and the leaves were detected by PCR with specific primers to screen the positive transgenic T1 lines. The screening steps described above were repeated to detect the transgenic T2 seedlings. The T3 seeds were harvested and used to find out the lines with 100% survives from herbicide, which was considered as the homozygote lines of T3 generation. The T3 homozygote lines were further used to conduct qRT-PCR analysis. Based on this, two over-expression lines of transgenic Arabidopsis were selected for further studies. For Al tolerance experiments, seeds of wild type and two overexpression lines for each gene were germinated on solid MS plates in conditions of 16 h light/8 h dark under 23 °C. Four days after germination, the initial root lengths of uniform seedlings were measured, and the seedlings were subsequently transplanted to 1/5 Hoagland nutrient solution (pH 4.5, without H_2_PO_4_, with 0.5 mM CaCl_2_) with or without 5 µM AlCl_3_ for 48 h. The root length before and after Al treatment was measured to calculate the root elongation.

For the functional characterization of *GmNTL4*, the uniform seedlings of wild type and *GmNTL4*-overexpressing lines were selected and treated with 5 μM AlCl_3_ for 24, 48, and 72 h. The root elongation was measured as mentioned above.

### 4.6. RNA-seq Transcriptomic Analysis

For RNA-seq transcriptomic analysis, uniform seedlings of both wild type and *GmNTL4*-overexpressing Arabidopsis were grown in modified 1/5 Hoagland nutrient solution (pH 4.5, without H_2_PO_4_^−^, with 0.5 mM CaCl_2_) with 5 µM AlCl_3_ for 48 h [22]. The roots of each genotype were separately harvested for RNA-seq transcriptomic analysis. The mRNA purification, library preparation, and sequencing were conducted as described before [69]. Briefly, the qualified libraries were sequenced using the Illumina platform with the sequencing strategy of PE150. The raw data was filtered to obtain high-quality clean data for subsequent analyses. The reference genome (*Arabidopsis thaliana* TAIR10) and the annotation file were downloaded from ENSEMBL database (http://www.ensembl.org/index.html, accessed on 12 May 2021). HISAT2 was used to align the clean data to the reference genome. The mapping result was viewed using the integrative genomics viewer (IGV). Reads count for each gene was counted by HTSeq, and fragments per kilobase million mapped reads (FPKM) were used to estimate the gene expression level. DEGseq was used for differential gene expression analysis, and genes with q ≤ 0.05 and |log2_ratio| ≥ 1 were identified as differentially expressed genes (DEGs). The DEGs significantly enriched biological functions and pathways were analyzed through the gene ontology (GO) database (http://geneontology.org/, accessed on 1 June 2021) and Kyoto encyclopedia of genes and genomes (KEGG) database (http://www.kegg.jp/, 6 June 2021), respectively. The RNA-seq data were uploaded to the gene expression omnibus (GEO) on the National Center for Biotechnology Information search database (NCBI) (https://www.ncbi.nlm.nih.gov/geo/query, accessed on 27 July 2021) with the accession number of GSE180839.

### 4.7. Statistical Analysis

All data were analyzed by Student’s t-tests using Microsoft Excel 2010 (Microsoft company, Seattle, WA, USA) and SPSS 13.0 (SPSS Institute, Chicago, IL, USA).

## Figures and Tables

**Figure 1 ijms-22-12854-f001:**
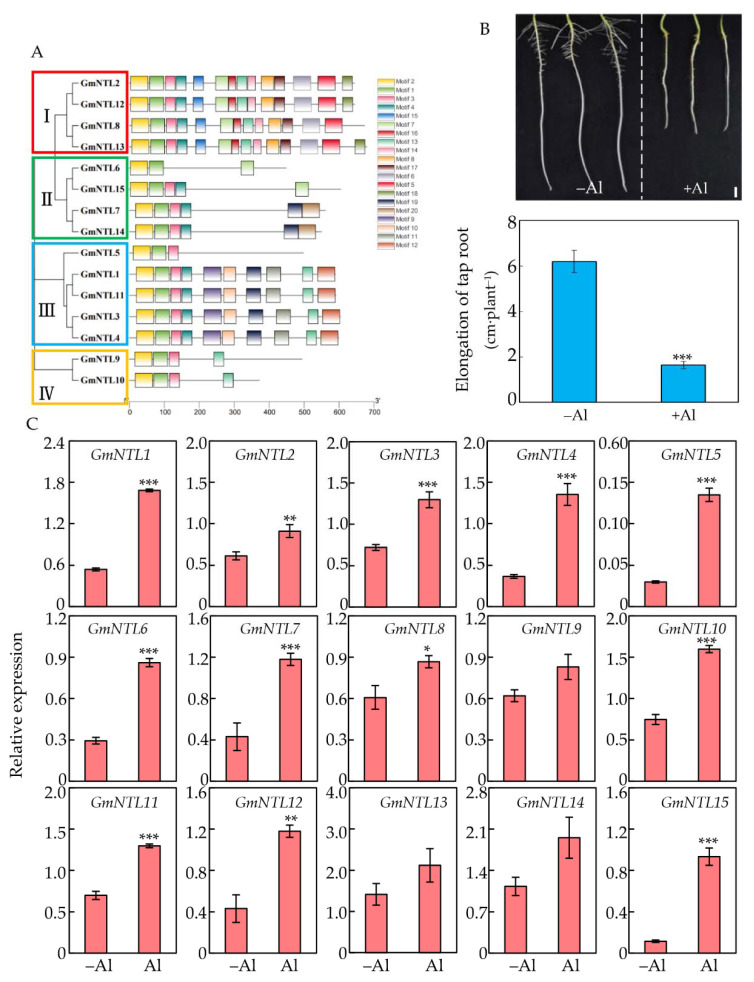
Expression of *GmNTL* members in response to Al toxicity. (**A**) Phylogenetic tree and conserved protein motifs in GmNTLs. The phylogenetic tree was constructed based on the amino acid sequences of GmNTLs using MEGA 7.0.14 software. The four clades were shown in different colors; (**B**) Phenotype and taproot elongation of soybean seedlings treated with (+Al) or without (−Al) 50 µM AlCl3 for 48 h; (**C**) Relative expression of *GmNTLs* in soybean roots tips (0–2 cm) in response to Al. Asterisks indicate significant differences between the +Al treatment and −Al control (*: 0.01 < *p* < 0.05; **: 0.001 < *p* < 0.01; ***: *p* < 0.001).

**Figure 2 ijms-22-12854-f002:**
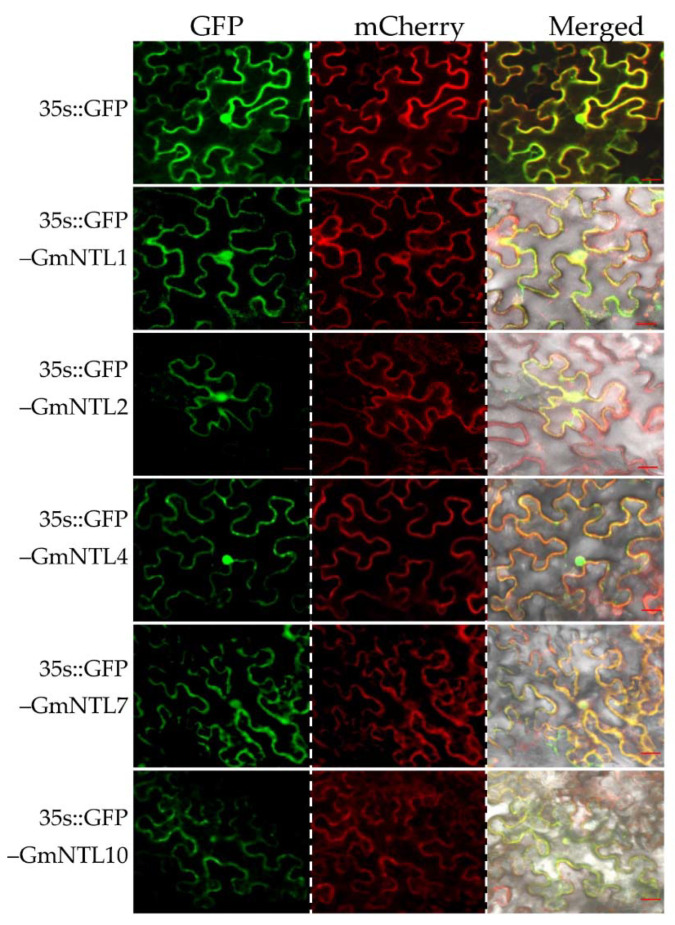
Subcellular localization analysis of GmNTLs in tobacco leaves. The green fluorescence derived from either 35S::GFP and 35S::GFP-GmNTLs, and the red fluorescence derived from 35S:AtPIP2A-mCherry were observed by the confocal microscope. Bar = 20 µm.

**Figure 3 ijms-22-12854-f003:**
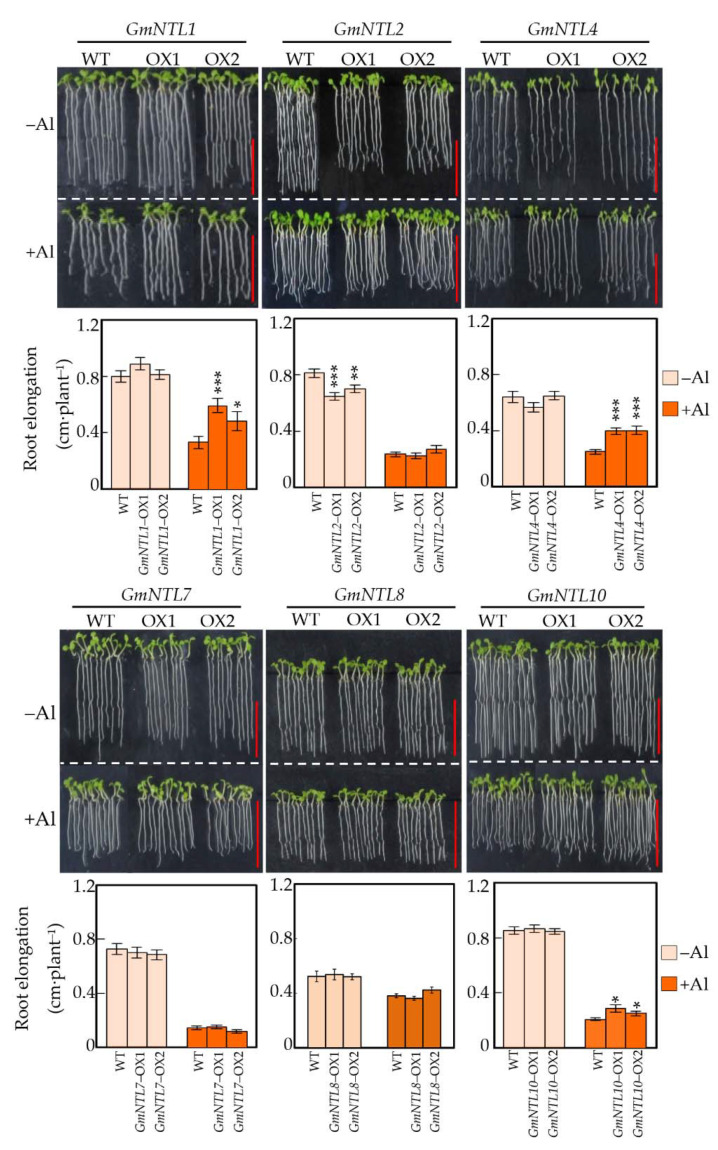
Effects of overexpressing *GmNTLs* on Arabidopsis primary root elongation in response to Al toxicity. Uniform seedlings of wild-type (WT) and transgenic seedlings (OX) with root lengths of 1 cm were treated with or without 5 µM Al for 48 h. Each bar represents the mean of four biological replicates with standard error. Asterisks indicate significant differences between WT and OX (*: 0.01 < *p* < 0.05; **: 0.001 < *p* < 0.01; ***: *p* < 0.001). Bar = 0.5 cm.

**Figure 4 ijms-22-12854-f004:**
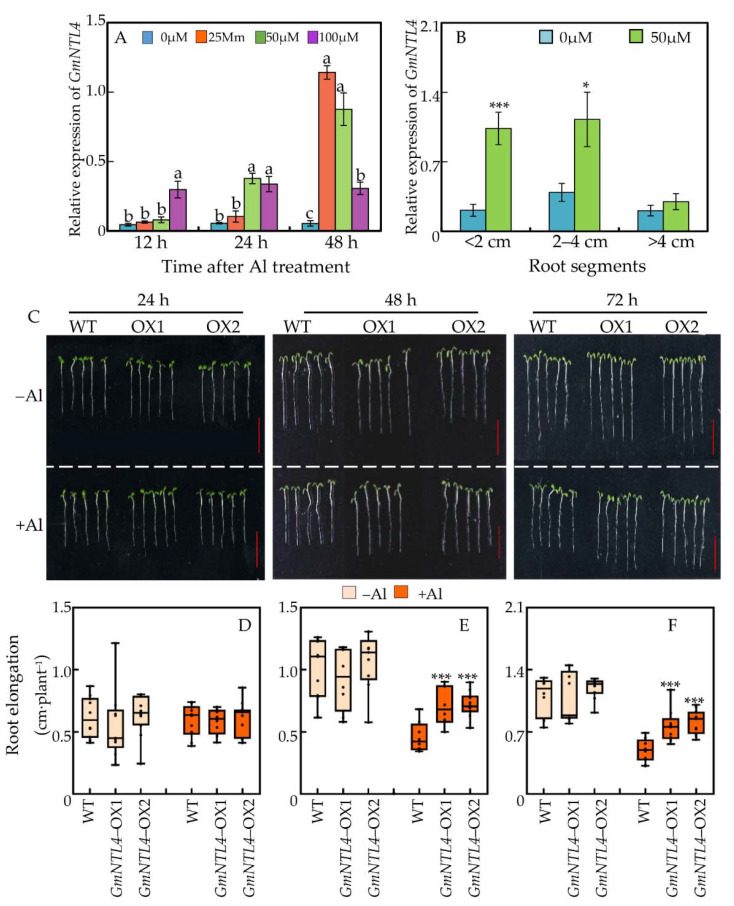
Functional characterization of *GmNTL4*. (**A**) Expression pattern of *GmNTL4* in response to Al availability and treatment time. Lowercase letters represent significantly different groups in same treatment time; (**B**) Al-regulated expression pattern of *GmNTL4* in different root segments. Asterisks indicate significant differences between 0 µM and 50 µM Al treatment in same root segments (*: *p* < 0.05 and *** *p*: < 0.001); (**C**) Phenotype of transgenic Arabidopsis with *GmNTL4* overexpressing in response to Al toxicity. Bar = 0.5 cm; (**D**–**F**) Effects of *GmNTL4* overexpression on the root elongation of Arabidopsis after 24 h (**D**), 48 h (**E**), and 72 h (**F**) of Al treatment. Asterisks indicate significant differences between WT and OX lines (***: *p* < 0.001).

**Figure 5 ijms-22-12854-f005:**
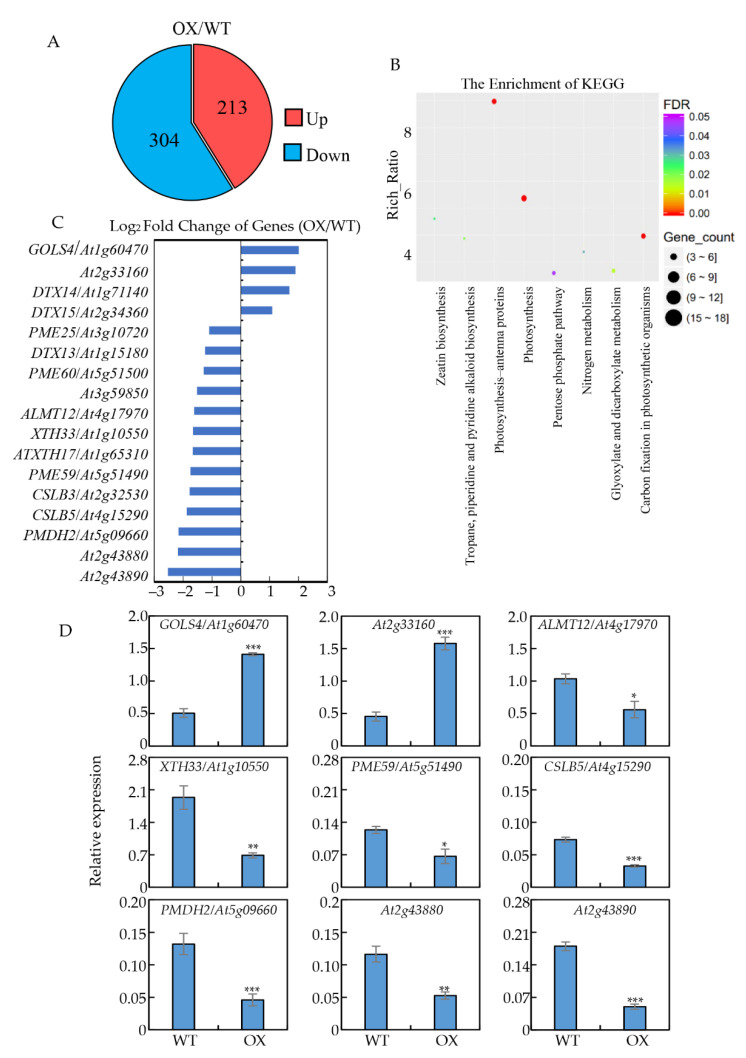
Gene expression profiles regulated by GmNTL4 in Arabidopsis. (**A**) Venn diagram showing the overlap of differentially regulated genes in *GmNTL4*-overexpressing plants compared to wild-type; (**B**) KEGG analysis of commonly regulated genes by GmNTL4 in response to Al; (**C**) List of partial GmNTL4-regulated genes in response to Al from RNA-seq; (**D**) qRT-PCR analysis of transcription accumulation of several genes in wild type and *GmNTL4*-overexpressing plants. Asterisks indicate significant differences between WT and OX (*: 0.01 < *p* < 0.05; **: 0.001 < *p* < 0.01; ***: *p* < 0.001).

## Data Availability

The original contributions presented in the study are publicly available. This RAN-seq raw data can be found on the NCBI repository, accession number: GSE180839.

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
