# Peer review of "Functional Characterization of Aluminum (Al)-Responsive Membrane-Bound NAC Transcription Factors in Soybean Roots"

_ijms, 2021, doi:10.3390/ijms222312854_

Round 1

Reviewer 1 Report

The article entitled "Functional Characterization of Al-responsive 2 Membrane-Bound NAC Transcription Factors in Soybean" is well planned, executed and overall it is a nice study.  However, It needs minor revision before accepted for publication.

Minor comments
Q1# The introduction should must be include the effect of Alumninum toxicity towards soybean crops. And which reason the study was plannned? What is the major challenges with Al for soybean cultivation.

Q2# Line 60-70 should be crisp.

Q3# At line 87; Author state that, The membrane-bound NAC transcription factors (NTLs) is one of the subfamily of NAC super-family. put the refrence Singh, S., et al. "The biotechnological importance of the plant-specific NAC transcription factor family in crop improvement." Journal of Plant Research (TSI) 134.3 (2021): 475-495.

Q4# Author must include which Taq Master mix used for qRT-PCR and waht was the cycle condition of qRT PCR? under section 4.3 how they calculated the relative fold change or put the refernces.

Q5# Author must include how they screen transgenic seeds or T1 line under section 4.5.

Q6# In figure 5 c and d must be included short name for the gene along with ID or only gene name, so that reader can easily understand.

Author Response

Response to Reviewer 1 Comments

Point 1: The introduction should must be include the effect of Alumninum toxicity towards soybean crops. And which reason the study was plannned? What is the major challenges with Al for soybean cultivation.

Response 1: Thank you for your valuable suggestions. We have added the information in the introduction. Please check it in the revised manuscript in line 97 to 103.

Point 2: Line 60-70 should be crisp.

Response 2: Thank you very much. We have revised it accordingly. Please check it in details in the revised manuscript in line 53 to 58.

Point 3: At line 87; Author state that, The membrane-bound NAC transcription factors (NTLs) is one of the subfamily of NAC super-family. put the refrence Singh, S., et al. "The biotechnological importance of the plant-specific NAC transcription factor family in crop improvement." Journal of Plant Research (TSI) 134.3 (2021): 475-495.

Response 3: Thank you very much. We have added the reference, please check this in details in the revised introduction and the reference in line 74 to 74.

Point 4: Author must include which Taq Master mix used for qRT-PCR and waht was the cycle condition of qRT PCR? under section 4.3 how they calculated the relative fold change or put the references.

Response 4: Thank you for your helpful suggestions. We have revised it accordingly. Please check this in details in the revised materials and methods section 4.3.

Point 5: Author must include how they screen transgenic seeds or T1 line under section 4.5.

Response 5: Thank you very much for your helpful suggestions. We have described the details in the revised materials and methods section 4.5.

Point 6:  In figure 5 c and d must be included short name for the gene along with ID or only gene name, so that reader can easily understand.

Response 6: Thank you very much for your careful and valuable suggestions. We have revised it accordingly. Please check it in the revised figure 5.

Reviewer 2 Report

Comments to authors

MS#ijms-1449244

The research work entitled” Functional Characterization of Al-responsive Membrane-Bound NAC Transcription Factors in Soybean” has been performed. In this study, the authors characterized the Aluminum (Al)-induced NAC Transcription Factors in soybean roots. The style of data presentation, interpretation of research gap in introduction section, and discussion related to this study could have been better compared to present format of this manuscript. Anyway, the authors are requested to revise the following points, which are as follows:

Major comments:

  1. Several grammatical and structural errors observed in this manuscript. Therefore, language of the entire manuscript should be checked by a language expert in a similar field of experience.
  2. Research gap: the author largely described the Al-responsive mechanism of plants (lines 33-82). This section should be reduced. And should be added an section on “ how molecular characterization of genes/transcription factors would be effective tool for plant improvements”. Some update references should be added (e.g. DOI: 1111/pce.13676 ; https://doi.org/10.3389/fpls.2017.00073 so on).
  3. Dose of Al and treatment duration and phenotypic changes: The author used 50 µM AlCl3 in 0.5 mM CaCl2 solution (pH 4.2) for 24 h (lines 529-530). Al is less Al competed to other toxic metal like Cd, As, Cr, Pb, and Hg. In acidic soils (pH < 5.0), phototoxic-aluminum (Al3+) rapidly inhibits root growth, and subsequently affects water and nutrient uptake in plants. Al toxicity highly dependable on medium pH and its solubility. Plant growth and other morphological changes occurred very slowly compared to other toxic metal in my experience. So, I am afraid and wondering to see the massive phonotypical difference within only 24 h (Fig.1B, line 181). The author should check the treatment dose and duration of treatment.
  4. The author should clarify the gene naming information that it essential to ensure gene function. Furthermore, the authors are requested to provide all the gene accession number for supplementary Table S2.

  1. Old references have been cited in introduction and result-discussion sections: Out of 74 references, total 39 (52.70 %) are very old which were taken from almost 10 years before. So, the authors are requested to replace very old reference by new ones. Specially, in introduction and discussion sections.

  1. Lack of correlations observed in discussion section. A good discussion contains-i) Principles and relationship which can be supported by the results; ii) emphasis on results and conclusions that agree and disagree with other work(s); iii) should be avoided the repetition of the results with figure number in discussion sections (lines 89-90); iii) theoretical implications or short summary/findings/suggestions of each section(s) where applicable. Therefore, the author should follow these above points to improve this section.

Minor comments:

  1. In Title: “Al” should be elaborated within bracket “Aluminum (Al)”, and the studied plants part “Roots” should be added with “Soybean Roots”. Please revise asFunctional Characterization of Aluminum (Al)-Responsive Membrane-Bound NAC Transcription Factors in Soybean Roots”.
  2. Line 17: “However, whether NTLs are involved in plant responses to aluminum (Al) toxicity remain uncharacterized”. This line should be revised as ”This study was aimed to functional characterization of soybean membrane bound NAC transcription factors in response to Al-toxicity, which is largely characterized.”
  3. Lines 46: all gene names should be italic throughout the manuscript.
  4. Introduction section: a consequence should be maintained among the sections.
  5. Difference in root elongation for Fig. 3: should be added in results and their interpretation must be added description section with update reference(s).
  6. Has the author performed (ICP-MS) experiments between –Al and Al+ lines/plants? If so, the addition of this data it would be exciting.
  7. Conclusion section: I did not find any conclusion section. A short discussion is appreciable, please add this section. Additionally, and the overall findings, consequence and a clear suggestion(s) can be provided in conclusion section.

Author Response

Response to Reviewer 2 Comments

The research work entitled” Functional Characterization of Al-responsive Membrane-Bound NAC Transcription Factors in Soybean” has been performed. In this study, the authors characterized the Aluminum (Al)-induced NAC Transcription Factors in soybean roots. The style of data presentation, interpretation of research gap in introduction section, and discussion related to this study could have been better compared to present format of this manuscript. Anyway, the authors are requested to revise the following points, which are as follows:

Major comments:

Point 1: Several grammatical and structural errors observed in this manuscript. Therefore, language of the entire manuscript should be checked by a language expert in a similar field of experience.

Response 1: Thank you very much for pointing out the grammatical and structural errors in our manuscript. We have carefully checked over and corrected the grammatical and structural errors. Please check it in the revised manuscript.

Point 2: Research gap: the author largely described the Al-responsive mechanism of plants (lines 33-82). This section should be reduced. And should be added an section on “how molecular characterization of genes/transcription factors would be effective tool for plant improvements”. Some update references should be added (e.g. DOI: 1111/pce.13676 ; https://doi.org/10.3389/fpls.2017.00073 so on).

Response 2: Thank you very much for your valuable suggestion. The manuscript has been revised accordingly and the references have been added. Please check it in details in the introduction section from line 71 to 75.

Point 3: Dose of Al and treatment duration and phenotypic changes: The author used 50 µM AlCl3 in 0.5 mM CaCl2 solution (pH 4.2) for 24 h (lines 529-530). Al is less Al competed to other toxic metal like Cd, As, Cr, Pb, and Hg. In acidic soils (pH < 5.0), phototoxic-aluminum (Al3+) rapidly inhibits root growth, and subsequently affects water and nutrient uptake in plants. Al toxicity highly dependable on medium pH and its solubility. Plant growth and other morphological changes occurred very slowly compared to other toxic metal in my experience. So, I am afraid and wondering to see the massive phonotypical difference within only 24 h (Fig.1B, line 181). The author should check the treatment dose and duration of treatment.

Response 3: Thank you very much for your carefulness and the valuable suggestions. We have double checked our data and the methods. And it is 48 h instead of 24 h. The manuscript has been revised accordingly.

Point 4: The author should clarify the gene naming information that it essential to ensure gene function. Furthermore, the authors are requested to provide all the gene accession number for supplementary Table S2.

Response 4: Thank you very much for the very helpful suggestions. We have added the information in Table S2 and Figure 5.

Point 5:  Old references have been cited in introduction and result-discussion sections: Out of 74 references, total 39 (52.70 %) are very old which were taken from almost 10 years before. So, the authors are requested to replace very old reference by new ones. Specially, in introduction and discussion sections.

Response 5:  Thank you very much for your useful suggestions. We have deleted some old references and added some new ones. Please check it in details in the revised manuscript and references.

Point 6: Lack of correlations observed in discussion section. A good discussion contains-i) Principles and relationship which can be supported by the results; ii) emphasis on results and conclusions that agree and disagree with other work(s); iii) should be avoided the repetition of the results with figure number in discussion sections (lines 89-90); iii) theoretical implications or short summary/findings/suggestions of each section(s) where applicable. Therefore, the author should follow these above points to improve this section.

Response 6: Thank you very much for your valuable comments. We have revised the discussion accordingly. Please check the details in the revised discussion section.

Minor comments:

Point 7: In Title: “Al” should be elaborated within bracket “Aluminum (Al)”, and the studied plants part “Roots” should be added with “Soybean Roots”. Please revise as “Functional Characterization of Aluminum (Al)-Responsive Membrane-Bound NAC Transcription Factors in Soybean Roots”.

Response 7: Thank you very much for the very helpful suggestion. We have revised the title accordingly.

Point 8: Line 17: “However, whether NTLs are involved in plant responses to aluminum (Al) toxicity remain uncharacterized”. This line should be revised as” This study was aimed to functional characterization of soybean membrane bound NAC transcription factors in response to Al-toxicity, which is largely characterized.”

Response 8: Thank you very much for your valuable suggestions. We have revised it accordingly. Please check it in details in the revised abstract.

Point 9: Lines 46: all gene names should be italic throughout the manuscript.

Response 9: Thank you very much for the helpful suggestion. We have revised them accordingly. Please find it in details in the revised manuscript.

Point 10: Introduction section: a consequence should be maintained among the sections.

Response 10: Thank you very much for your valuable suggestion. The information has been added to the introduction. Please check it in details in the revised manuscript.

Point 11: Difference in root elongation for Fig. 3: should be added in results and their interpretation must be added description section with update reference(s).

Response 11: Thank you for your valuable suggestions. We have revised accordingly. Please check it in details in the result section 2.3 and discussion section.

Point 12: Has the author performed (ICP-MS) experiments between –Al and Al+ lines/plants? If so, the addition of this data it would be exciting.

Response 12: Thank you very much for your valuable comments. you are right it would be very exciting to see the data about Al content in both cell walls and cell saps in the transgenic Arabidopsis. But it is sorry that we did not perform (ICP-MS) experiments due to the poor facilities.

Point 13: Conclusion section: I did not find any conclusion section. A short discussion is appreciable, please add this section. Additionally, and the overall findings, consequence and a clear suggestion(s) can be provided in conclusion section.

Response 13: Thank you very much for the suggestion. The conclusion section was in the last paragraph of the manuscript. We have made some minor revision accordingly. Please check it in details in the revised manuscript.

Round 2

Reviewer 2 Report

Comments to authors

#MS-ijms-1449244-R1

I checked the revision of the manuscript entitled "Functional Characterization of Al-responsive Membrane-Bound NAC Transcription Factors in Soybean"

The author revised all of the points raised by the reviewer previously.
Despite the fact that one requested reference was not added and interpreted in the introduction section (doi.org/10.1111/pce.13676).
Furthermore, it would be ideal if the author could revise the language through the manuscript, as large changes in the text appear vegue.

Please revise the above queries before it further processing.

Author Response

I checked the revision of the manuscript entitled "Functional Characterization of Al-responsive Membrane-Bound NAC Transcription Factors in Soybean"

Point 1: The author revised all of the points raised by the reviewer previously.
Despite the fact that one requested reference was not added and interpreted in the introduction section (doi.org/10.1111/pce.13676).

Response 1: Thank you very much for your carefulness and the helpful suggestion. We have added the reference in the revised manuscript. Please check this in details in the revised introduction and the reference section.

Point 2: Furthermore, it would be ideal if the author could revise the language through the manuscript, as large changes in the text appear vegue. Please revise the above queries before it further processing.

Response 2: Thank you very much for the helpful suggestion. We have check it again throughout the manuscript and made the language revisions, which are marked up using the “Track Changes”. Please find it in details in the revised manuscript with “Track Changes”.